# Assessing Differences in Attitudes toward Occupational Safety and Health Measures for Infection Control between Office and Assembly Line Employees during the COVID-19 Pandemic in Germany: A Cross-Sectional Analysis of Baseline Data from a Repeated Employee Survey

**DOI:** 10.3390/ijerph20010614

**Published:** 2022-12-29

**Authors:** Jana Soeder, Anna T. Neunhöffer, Anke Wagner, Christine Preiser, Benjamin Rebholz, Diego Montano, Norbert Schmitz, Johanna Kauderer, Falko Papenfuss, Antje Klink, Karina Alsyte, Monika A. Rieger, Esther Rind

**Affiliations:** 1Institute of Occupational and Social Medicine and Health Services Research, University Hospital Tübingen, University Tübingen, Wilhelmstraße 27, 72074 Tübingen, Germany; 2Department of Population-Based Medicine, Institute of Health Sciences, University Hospital Tübingen, Hoppe-Seyler-Straße 9, 72076 Tübingen, Germany; 3Medical Services, Robert Bosch GmbH, P.O. Box 10 60 50, 70049 Stuttgart, Germany

**Keywords:** COVID-19 pandemic, workforce, occupational safety and health, workplace health, infection control measures, occupational SARS-CoV-2 risk of infection, baseline data, working conditions

## Abstract

In our study, we investigated possible differences across occupational groups regarding employees’ perceived work-related risk of infection with SARS-CoV-2, attitudes toward technical, organisational, and personal occupational safety and health (OSH) measures for infection control, and factors associated with this attitude. We analysed baseline data (10 August to 25 October 2020) from a repeated standardised online survey distributed at a worldwide leading global supplier of technology and services in Germany. 2144 employees (32.4% women; age (mean ± SD): 44 ± 11 years) who worked predominantly remotely (*n* = 358), at an on-site office (*n* = 1451), and assembly line/manufacturing (*n* = 335) were included. The work-related SARS-CoV-2 risk of infection differed between office employees working remotely and on-site (mean ± SD = 2.9 ± 1.5 vs. 3.2 ± 1.5; Mann-Whitney-U-Test: W = 283,346; *p* < 0.002; ε^2^ = 0.01) and between on-site office and assembly line/manufacturing employees (3.8 ± 1.7; W = 289,174; *p* < 0.001; ε^2^ = 0.02). Attitude scores toward technical OSH-measures differed between remote and on-site office (4.3 ± 0.5 vs. 4.1 ± 0.6; W = 216,787; *p* < 0.001; ε^2^ = 0.01), and between on-site office and assembly line/manufacturing employees (3.6 ± 0.9; W = 149,881; *p* < 0.001; ε^2^ = 0.07). Findings were similar for organisational and personal measures. Affective risk perception, COVID-19-specific resilience, and information about COVID-19-related risks were associated with the employees’ attitudes. To promote positive attitudes, it seems to be important to consider occupational-group-specific context factors when implementing OSH-measures for infection control.

## 1. Introduction

The severe acute respiratory syndrome Coronavirus 2 (SARS-CoV-2) has evolved rapidly since the end of 2019. On 30 January 2020, the World Health Organisation (WHO) declared the communicable coronavirus disease (COVID-19) a Public Health Emergency of International Concern [1], and then, on 11 March 2020, a pandemic [2]. Due to a lack of effective treatment and vaccines at that time, governments worldwide introduced temporary preventive behavioural measures to slow down the spread of the virus, prevent overload on local health systems, and enable the identification of common sources of exposure to trace infection chains [3,4].

Prior research has shown that the workplace is a setting with the potential to reinforce or mitigate the spread of SARS-CoV-2 depending on workplace characteristics [5,6,7]. Even during previous influenza pandemics, occupational groups working in crowded workplaces faced a high risk of exposure [8]. The reduction of direct contact at work was identified as an effective measure to limit the transmission of various influenza virus strains [8]. Concerning the risk assessment of work-related exposure to SARS-CoV-2, the WHO has established definitions for three occupational groups: (1) “low exposure risk”: employees without constant or close contact with clients or colleagues; (2) “medium exposure risk”: employees fulfilling job tasks requiring close and constant contact with the clients or colleagues; (3) “high exposure risk”: employees caring for SARS-CoV-2-infected patients or with close contact with individuals suspected of being infected [9]. Accordingly, in the nonhealthcare sector, the risks of work-related exposure to SARS-CoV-2 differ greatly, for example, between office workers (low) and skilled workers on the assembly line or in production (medium). Besides preventive behavioural measures, the COVID-19 pandemic required the implementation of structural measures to combat infection in the workplace and protect the occupational safety and health (OSH) of professionals across all occupational settings according to their exposure risk [7,10,11].

In Germany, the German Federal Ministry of Labour and Social Affairs (BMAS) provided a SARS-CoV-2 Occupational Safety and Health Standard in April 2020 to reduce the work-related risk of infection of the employees [12]. All companies in Germany were required to implement infection control measures regarding workplace setting, ventilation, remote work, working equipment, and hygiene rules on technical, organisational, and personal levels (TOP) [13,14]. The more extensive SARS-CoV-2 Occupational Health and Safety Regulation was introduced on 20 August 2020 [14] and complements the German national government’s pandemic roadmap in the private and public sectors. This roadmap provided recommendations on the targeting and effectiveness of public containment measures to serve as a toolbox for implementation by the single federal state governments according to the respective epidemiological situation [15]. Preventive behaviours to reduce virus transmission in private and public settings were also further promoted, e.g., maintaining distance, observing hygiene rules such as regular hand washing, mouth and nose protection, using the Corona-Warn-App, and room ventilation (DHM-AA or AHA+A+L formula) [16,17]. 

Attitudes affect behaviour: a high positive attitude toward a certain behaviour increases the probability of transferring plans into action [18], e.g., realising compliant preventive behaviour during the COVID-19 pandemic [19,20]. An attitude can be defined as “a psychological tendency that is expressed by evaluating a particular entity with some degree of favor or disfavor” [21]. The attitude’s formation is described as an ongoing process which occurs whenever new information is received related to the attitude object [18]. Therefore, attitude and attitude strength can change [18,22]. Previous research has shown that the attitude toward measures for infection control in public life (e.g., in the general Spanish population [23]) and in the workplace (e.g., in dentists [24]) influences the implementation and realisation of preventive behaviours. The German serial cross-sectional COVID-19 Snapshot Monitoring study (COSMO) has monitored different COVID-19-related topics, e.g., protective or preparedness behaviours and public risk perceptions, and used a representative quota sample for the German general population [25,26]. Overall, little is known about how employees of different occupational groups, such as office or assembly line/manufacturing workers, perceive their work-related risk of infection. Furthermore, preventive OSH-measures introduced for infection control elicited compulsive changes regarding working conditions, e.g., workplace environment and organisation [27]. There is a lack of research on the employees’ reactance towards these changes. A better understanding of how those employees experience their ‘new normal’, could therefore help to adapt these measures according to target group-specific characteristics and promote acceptance [10,28,29]. The personal relevance of measures influences attitude [30], e.g., previous research has shown that concern about the virus increases when personal contact with positive COVID-19 cases increases [19,31]. We therefore assume that a person who is personally affected by the OSH-measures, e.g., working on-site, might evaluate the consequences of the measures differently than a person who is not directly affected, e.g., working remotely.

Our objective is to compare the different work-related risks of exposure to SARS-CoV-2, based on the classification of the WHO [9], for employees who do office work predominantly remotely (at least 80% of the working time remote) with employees who do office work on-site with assembly line/manufacturing employees on the topics of:a)How did employees rate their risk of infection with SARS-CoV-2 in the summer and fall of 2020?b)What are the employees’ attitudes and reactances toward SARS-CoV-2 infection control measures in the workplace?c)What factors are associated with their attitudes toward OSH-measures for infection control?

## 2. Materials and Methods

### 2.1. Study Design

This cross-sectional study was conducted using baseline data from an employee survey that was administered three times. It is part of an explorative modular mixed-methods study project investigating how companies and employees in Germany dealt with adjusted working conditions due to infection control measures introduced during the COVID-19 pandemic [32,33]. This study was approved by the ethics committee of the Medical Faculty, University of Tübingen, and University Hospital of Tübingen in June 2020 (No.: 423/2020BO). The manuscript was reported following the STROBE checklist [34], see Appendix A.

### 2.2. Study Setting and Recruitment

The study was conducted at six German company sites of a worldwide leading global supplier of technology and services located in the federal states of Bavaria, Baden-Wurttemberg, and Lower Saxony. Baseline data were collected via a partly standardised online survey between 10 August and 25 October 2020. We distributed the invitations for all employees in close cooperation with the company’s corporate communications department by e-mail, newsletter, intranet, postcards, and posters including a link and QR code to the survey. The completion of the survey was voluntary, took about 25 min, and could be completed during working hours.

Regarding the epidemiological context at the time of data collection, the German national government and federal states faced the challenge of balancing constraints in private and public life such as contact restrictions, quarantine regulations for people with suspected or confirmed SARS-CoV-2-infection, and closure of educational or non-systematically relevant public and commercial institutions [15]. The 7-day incidence at the state level was used as a key indicator for the introduction or repeal of measures. During summer and until the end of September 2020, the 7-day incidence in Germany was below 20 cases per 100,000 inhabitants [35]. Until 31 October 2020, the 7-day incidence increased to 110 cases per 100,000 inhabitants [35]. Again, temporary restrictions (e.g., contact restrictions or restaurant closures) were discussed [3,15].

### 2.3. Study Population

The study population consists of professionals employed in the company. The following inclusion criteria were applied: minimum age of 18 years and the ability to access and complete the online survey in German. All included study participants provided informed consent. In this study, we focus on the two largest areas of work in this company. Employees working in the ‘office’ mainly carry out business activities at visual display units in separated office spaces of various sizes, including individual to open-plan offices or conference rooms. ‘Assembly line and manufacturing’ characterises hands-on factory or quality-control tasks on production parts performed on site at assembly lines, machines, laboratories, and cleanrooms.

### 2.4. Measures (Questionnaire)

The survey was developed in close interdisciplinary cooperation with professionals providing expert knowledge on health science research, occupational medicine, and corporate communication. Our target group was involved in the design, choice of outcome measures, and recruitment of our research. Our questionnaire was pretested by eight academic volunteers and four target group representatives. Due to the short time between the onset of the pandemic and the first wave of data collection, the questionnaire was solely developed in the German language and conducted online via the survey tool Unipark provided by Questback AS [36]. In addition to self-developed items and those from prevailing surveys [37], e.g., COSMO [26], we included validated scales, e.g., the Copenhagen Psychosocial Questionnaire (COPSOQ) [38], and the short scale Social Desirability-Gamma (KSE-G) [39]. See Appendix A for further details and sources [40,41,42,43,44,45,46,47].

Figure 1 shows the main subject areas of the online employee survey:I)socio-demographics (e.g., age, gender);II)workplace characteristics (e.g., professional activity, performing shift work);III)the employee’s perception of SARS-CoV-2 and COVID-19 in general (e.g., affective risk perception describing fear and worry about the coronavirus);IV)the employee’s evaluation of work-related stress due to COVID-19-induced changes in working conditions elicited by introduced infection control measures (e.g., perceived probability of contracting COVID-19 in the workplace, primary workplace location during the pandemic);V)the employee’s attitudes toward SARS-CoV-2-related OSH-measures for infection control in the workplace (e.g., ‘protection of employees with plexiglas planes’; ‘decoupled break times‘).

**Figure 1 ijerph-20-00614-f001:**
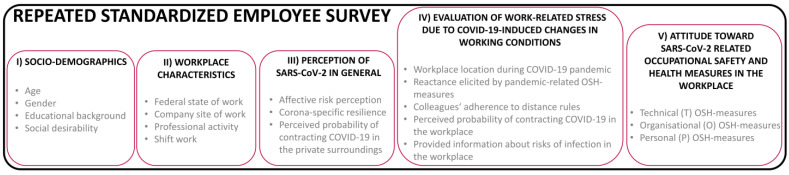
Main subject areas of the standardised online employee survey at baseline.

### 2.5. Outcomes

a)Perceived probability of contracting COVID-19 in the workplace: the subjectively perceived risk of infection with SARS-CoV-2 in the workplace was evaluated on a 7-point Likert scale ranging from ‘extremely unlikely’ to ‘extremely likely’ (1–7), similarly to Betsch et al. [48].b)Perceived probability of contracting COVID-19 in the private surroundings: the subjectively perceived risk of infection with SARS-CoV-2 in the private surroundings was evaluated on a 7-point Likert scale ranging from ‘extremely unlikely’ to ‘extremely likely’ (1–7), similarly to Betsch et al. [48].c)Attitude toward OSH-measures for infection control: at the company sites implemented OSH-measures were grouped according to the hierarchy of implementation according to the German SARS-CoV-2 Occupational Safety and Health Standard [12]: technical (T) (10 items), organisational (O) (6 items), and personal measures (P) (10 items), see Appendix A. Participants rated the perceived adequacy of each item along a 5-point Likert scale ranging from ‘not at all suitable’ to ‘very suitable’ (1–5). Mean scores for the technical, organisational, and personal measures were calculated for further statistical analysis. High values indicate a positive attitude. Cronbach’s alpha was computed to evaluate the internal consistency reliability (each score’s Cronbach’s α ≥ 0.77).d)Reactance toward implemented infection control measures in the workplace: similar to previous research [49,50], we asked participants to rate their feelings of frustration, annoyance, restriction of freedom, and disruption regarding implemented infection control measures explicitly in the workplace on a 7-point Likert scale from ‘not at all’ to ‘very much’ (1–7). A mean score of reactance was calculated for further statistical analysis (Cronbach’s α = 0.92). A high reactance score reflects the experience of an unpleasant emotional reaction triggered by the introduced OSH-measures in the workplace which is likely to reinforce the intention to regain restricted freedom [50].

Considered covariates included: age (complete years), gender (male, female, diverse), date of participation, nationality (German/non-German), educational background (categorised as primary, intermediate or high according to [51]), company site, involvement in shift work (yes/no), social desirability [39], trust in colleagues to adhere to distance rules (7-point Likert scale ranging from ‘not at all’ to ‘always’ (1–7)), affiliation to risk group for developing severe COVID-19 according to [52,53] (yes/no), provided information about potential risks of infection with SARS-CoV-2 in the workplace (5-point Likert scale ranging from ‘inadequately’ to ‘very good’ (1–5)), affective risk perception (‘very low’ to ‘very high’ (1–7), mean score across three items [26,54], Cronbach’s α = 0.80), COVID-19-specific resilience describing the ability to cope with difficulties caused by the COVID-19 pandemic (‘very low’ to ‘very high’ (1–7), mean score across four items [26], Cronbach’s α = 0.70), see Appendix A.

### 2.6. Statistical Analysis

We described the baseline characteristics using R [55,56]. Descriptive results were reported as means ± SDs with interquartile ranges (IQR) for continuous measures including Likert scales and as percentages for categorical variables. Participants of diverse gender were not analysed separately due to the low number. Mean calculation for score computation was performed for previously validated scales according to their manual and for self-developed scores, e.g., attitude toward OSH-measures, according to the mean-across-available-item approach [57]. Non-parametric Kruskal-Wallis tests were performed to identify differences between the three occupational groups [58]. As we mainly focus on the differences between remote and on-site office employees, respectively between on-site office and assembly line/manufacturing employees, Mann-Whitney-U tests were further performed [58]. We report effect sizes as epsilon-squared (ε^2^), with ε^2^ < 0.083 representing a weak, ε^2^ < 0.268 a moderate, and ε^2^ > 0.268 a strong effect [59]. A two-sided *p*-value below 0.05 was considered statistically significant. For multivariate analysis, missing values of non-sociodemographic variables in included observations were replaced via predictive mean matching using the MICE package in R [60]. Results were similar when performing the analysis without observations with missing values, see Appendix A. The scores reflecting the attitudes toward OSH-measures for infection control in the workplace (dependent variable) were non-normally distributed due to a ceiling effect located at 5 (highest response option). The IQR of all respondents’ ratings on the attitude toward technical measures was 3.7 to 4.6 (skewness = −1.05). Therefore, the multivariate Tobit regression model was performed to identify factors associated with the attitude toward OSH-measures [61]. We controlled computed regression models for social desirability [39], 7-day incidence [62], educational background, and company site. Based on our survey modules, we investigated block-wise via forward selection to decide on the most reduced models for remote, on-site office, and assembly line/manufacturing employees [63]. Regression models for technical OSH-measures are presented in the text, see Appendix A for similar results on organisational and personal OSH-measures.

## 3. Results

### 3.1. Socio-Demographic Characteristics of Participants

The response rate was 22% and 2144 employees were included in the analysis, see Table 1. The CONSORT flowchart is attached as Appendix A.

### 3.2. Perceived Probability of Contracting COVID-19 in the Workplace

Figure 2 shows the perceived probability of contracting COVID-19 in the workplace for predominantly remotely working office employees compared with on-site working office employees and assembly line/manufacturing employees (Kruskal-Wallis-Test: Chi-squared = 49.828; *p* < 0.001). The risk perception differed between remote and on-site office employees (Mann-Whitney-U-Test: W = 283,346; *p* < 0.002; ε^2^ = 0.01) and between on-site office and assembly line/manufacturing employees (Mann-Whitney-U-Test: W = 289,174; *p* < 0.001; ε^2^ = 0.02): approximately one fifth of the remote office employees (17.9%) and of those working on-site (19.8%), and a third (33.3%) of the assembly line/manufacturing employees rated the perceived probability of contracting COVID-19 in the workplace as likely (≥5.0 on 7-point Likert scale), see Table 1. Figure 2 additionally shows the perceived probability of contracting COVID-19 in private surroundings (Kruskal-Wallis-Test: Chi-squared = 4.7752; *p* = 0.09).

Office employees working predominantly remotely perceived a higher risk perception in their private surroundings than in the workplace (W = 88,878; *p* < 0.001; ε^2^ = 0.12). Those office employees who worked on-site reported a higher risk perception in private surroundings than in the workplace (W = 1,299,570; *p* < 0.001; ε^2^ = 0.05). No difference was found for assembly line/manufacturing employees (W = 55,239; *p* = 0.722).

### 3.3. Attitudes and Reactances toward OSH-Measures for Infection Control

Figure 3 shows the employees’ attitudes toward technical (A) occupational infection control measures for all three occupational groups (Chi-squared = 156.43; *p* < 0.001), as well as toward organisational (B) (Chi-squared = 99.558; *p* < 0.001), and personal (C) measures (Chi-squared = 28.209; *p* < 0.001).

Mean scores for the attitudes were different between predominantly remotely and on-site working office employees toward technical (W = 216,787; *p* < 0.001; ε^2^ = 0.01) and organisational (W = 237,942; *p* = 0.034; ε^2^ = 0.00) but not for personal OSH-measures (W = 252,660; *p* = 0.615), see Table 1. Mean attitude scores toward technical (W = 149,881; *p* < 0.001; ε^2^ = 0.07), organisational (W = 163,125; *p* < 0.001; ε^2^ = 0.05), and personal OSH-measures (W = 198,262; *p* < 0.001; ε^2^ = 0.02) were also different between on-site working office employees and assembly line/manufacturing employees.

Reported levels of reactance toward changed working conditions elicited by introduced OSH-measures for infection control in the workplace were different for all three occupational groups, see Table 1 (Chi-squared = 48.56; *p* = 0.001). Office employees working predominantly remotely showed a lower level of reactance compared to on-site office employees (W = 76,905; *p* < 0.001; ε^2^ = 0.06). Further, on-site office employees showed a lower level of reactance than assembly line/manufacturing employees (W = 291,419; *p* < 0.001; ε^2^ = 0.02). 9.5% of the remote, 10.5% of the on-site office employees, and 20.8% of the employees working at the assembly line/in manufacturing reported a strong reactance (≥5.0 on a 7-point Likert scale).

### 3.4. Factors Associated with the Attitude toward Occupational Infection Control Measures

We found that a high COVID-19-specific resilience, high affective risk perception, having the feeling of being well-informed by the employer about potential COVID-19-health risks in the workplace, and a low reported level of reactance favour a positive attitude toward technical OSH-measures for infection control across all occupational groups, see Table 2. Associated with a positive attitude toward technical OSH-measures were the female gender for office workers and the male gender for assembly line/manufacturing workers. Trust in colleagues to adhere to distance rules showed a reinforcing effect when working on-site. Regarding attitudes toward organisational and personal OSH-measures, we found decreasing effects for gender but reinforcing effects for affective risk perception and reactance, see Appendix A.

## 4. Discussion

In the present cross-sectional analysis, we revealed a lower perceived risk of infection and a higher positive attitude toward OSH-measures for infection control among predominantly remotely working office employees in comparison to on-site working office employees and assembly line/manufacturing employees. Within the three investigated occupational groups, we identified gender, age, reactance, level of information about potential COVID-19-health risks, trust in colleagues to follow distance rules, COVID-19-specific resilience, and SARS-CoV-2 affective risk perception as influencing factors for OSH-measures for infection control.

### 4.1. Perceived Probability of Contracting COVID-19 in the Workplace

According to the WHO and the European Centre for Disease Prevention and Control [7,9], employees in indoor settings with close and frequent physical proximity to colleagues or customers face a higher risk of infection in the workplace. An online survey among company experts for safety and health protection among different German industries confirmed that the perceived risk should be considered differently with special regard to the number of contacts with colleagues or clients in the predominant workplace and working environment [64]. However, research with a focus on different occupational groups within a company is limited [7,11,64]. The results of this study show a lower risk perception in the workplace among predominantly remotely compared to on-site working office and assembly line/manufacturing employees. The fact that on-site workers perceive a higher risk of infection with SARS-CoV-2 in the workplace was previously explained by on-site workers fearing transmission to vulnerable friends or family members in addition to their own infection [19,65]. Within the German COSMO trend study, employed participants (13 October 2020; *N* = 944) rated their perceived probability of contracting COVID-19 in general as 3.7 on a 7-point Likert scale [66]. COSMO participants further indicated a high risk perception when meeting in groups or enclosed spaces in private or professional settings [67]. Compared to this general employed German population, our participants working in the office perceived a lower risk of infection in the workplace but a higher risk of infection in private surroundings. This did not apply to assembly line/manufacturing employees. Recent literature [68] has shown that personality traits and social patterns, e.g., eudaimonic and hedonic, influence attitudes toward infection control measures. With our data we could not investigate this any further.

Consistent with previous research [64,65,69], our findings therefore suggest target-group-specific OSH-measures for infection control under consideration of the workplace environment including, e.g., break rooms, canteens, and sanitary facilities.

### 4.2. Attitudes toward and Reactances Elicited by Occupational Infection Control Measures

A positive attitude toward preventive behaviour was previously shown to be associated with a higher probability of transferring those measures into action and acting according to COVID-19 policies [19,20]. We observed rather positive attitudes toward OSH-measures for infection control in our participants. It should be considered that successful safety cultures were shown to promote positive attitudes toward OSH-measures for infection control [70]. Before the COVID-19 pandemic, a concept of safety culture was established particularly in large companies or companies that are part of a corporate group [71,72]. Available human resources, communication structures or information channels, and high commitment of managerial staff to take responsibility enabled those companies to implement and adjust COVID-19 infection control measures rapidly [64,69,73]. It has been shown that even during the COVID 19 pandemic, the larger the organisation, the more likely it is to adopt flexible working conditions, such as remote working [74], and that frequent communication strategies promote compliance with infection control measures [23]. We recruited employees from a large company group already realising a clear concept of safety culture. A lack of research remains with similar studies conducted among companies of different sizes.

Regarding technical and organisational OSH-measures for infection control, the introduced measures elicited compulsive changes in the working environment and interaction with colleagues or leaders affecting primarily on-site employees’ daily work organisation. In our study population, predominantly remotely working employees reported higher positive attitudes compared to on-site employees. This is consistent with findings from a survey among German employees after the first pandemic winter in April 2021, where those spending at least half of their working time remotely rated the OSH-measures as more appropriate than those on-site [65]. A lower personal relevance of the measures could be the reason here [30]. For example, ventilating common rooms, cleaning equipment more frequently, or keeping a distance from colleagues do not directly impact work routines when working remotely. However, the shift toward working remotely where possible was highly recommended, e.g., by the federal governments, required rapid adaptation to the new working conditions, and led to a reinforced blurring between work and private life [29,75]. The ongoing shift from traditional work characteristics toward the ‘new normal’ via rapidly digitalised work processes has been boosted due to physical distancing at work to limit the number of new infections [64,69,76]. Regarding the reactance toward changed working conditions triggered by the implemented infection control measures, we observed lower levels of reactance among employees working remotely compared to on-site employees. Employees working at the assembly line/manufacturing reported higher levels of reactance, similar to the level of reactance toward public life restrictions in German employees (mean = 3.3; 27 October 2020; *n* = 927) [66]. It was previously reported that on-site working employees felt that they must invest a lot of energy at work while experiencing profound protective regulations in their daily work organisation [75,77]. They further experienced limited opportunities to receive social support from family and friends due to current pandemic-related government restrictions [75,77]. Establishing flexible work arrangements within the scope of possibilities was shown to solve work-life conflicts but is limited to jobs independent of shift work and manufacturing schedules [75]. In line with the existing literature [75], we found more positive attitudes and lower reactances toward workplace policies among on-site office employees compared to assembly line/manufacturing employees. We explain our findings with lower involvement in shift work, more flexible working arrangements, and easy access to formal internal information channels during working hours.

We identified an overall high positive attitude across the presently investigated occupational groups for personal OSH-measures. We found statistically significant but not practically relevant differences between the three groups (maximum difference of 0.3 points on a 5-point Likert scale). This might be due to the strong similarity of the personal measures to general behavioural policies in the private and public surroundings in Germany at that time, e.g., the DHM-AA formula [15,16].

### 4.3. Factors Associated with the Attitudes toward Work-Related Infection Control Measures

Within the three occupational groups, we found that low reactance, a high level of information about potential COVID-19-health risks, high trust in colleagues to follow distance rules, high COVID-19-specific resilience, and high SARS-CoV-2 affective risk perception promoted a positive attitude toward OSH-measures for infection control. This is consistent with previous findings [70,75,78]. Affective risk perception, as well as the amount of provided COVID-19-related health information, e.g., scientific facts about the pandemic or infection prevention, was shown to be strongly positively associated with engaging in preventive behaviours during the COVID-19 pandemic [78,79]. In contrast to the affective risk perception, the perceived probability of infection did not significantly influence attitude in our multivariate model. This is in line with the findings of a nationwide study in the Italian population, according to which affective risk perception seemed to explain protective behaviours better than the perceived probability of an infection with COVID-19 [80]. In the short-term, fear is the primary factor affecting attitudes toward infection control measures, but in the long run, communication that focuses on hope has been shown to be more effective in motivating adherence to OSH-measures for infection control and increasing persistence [81]. Knowledge about potential risks of infection in the workplace and understanding the need for preventive measures were also shown to reduce COVID-19-related stress factors and the feeling of frustration and restriction elicited by implemented OSH-measures [20,23,82,83]. Our findings confirm that the feeling of being well informed about potential health risks in the workplace by the employer strengthened attitudes toward implemented OSH-measures. Therefore, the employer can play an important role in providing trustworthy information, similar to public health education programs [23]. We can hypothesise that the reactances in assembly line/manufacturing employees were higher compared to office employees because they had limited access to digital communication channels and therefore received little information about potential work-related COVID-19-health risks during working hours.

A cross-sectional study revealed that good relationships with colleagues and leaders strengthened the positive perception of work [70], but in recent interviews with leaders, they reported that different experiences with the COVID-19 pandemic jeopardised team spirit [84]. Employers can support their employees in coping with pandemic-related challenges by maintaining, retaining, and protecting their resources. According to the Conservation-of-Resources Theory, providing this feeling of care might increase the employee’s attitude toward corporate COVID-19 measures [85]. Psychosocial resources, including resilience, self-efficacy, and optimism were shown to positively impact employees’ attitude toward their employers’ actions [85], which we can confirm for COVID-19-specific resilience. Age was previously shown to be positively associated with engaging in preventive behaviour, such as mask-wearing in Germany [78] or more frequent preventive behaviour by the elderly than by younger people [79]. The elderly and men were further reported to generally perceive a higher risk of COVID-19 for their own health, and for friends and family [86]. Within our sample, the explanatory power of age was low. For gender, we found that women working at the assembly line/manufacturing tended to perceive a more positive attitude toward technical OSH-measures than women. We did not find this effect for organisational or personal OSH-measures.

### 4.4. Strengths and Weaknesses

The strengths of the study include its large sample size and examination of different occupational groups. On the one hand, the online format of the survey ensured accessibility of the survey to employees working remotely due to the COVID-19 pandemic [87]. On the other hand, due to the online format we may have systematically excluded people who do not have time, resources, internet access, or email access at work, as they change their workplace frequently, e.g., different workplaces across company sites. Therefore, employees working at the assembly line/manufacturing were less likely to participate than office employees. We were only able to conduct the survey using the German language due to the limited time between the onset of the pandemic and data collection. A majority of participants reported not being at increased risk for developing severe COVID-19 according to medical criteria [52]. Our evaluable response rate is 22%, even though we put great effort in reaching all employees via mail as well as via printmedia, e.g., posters and postcards. In comparison to the overall workforce in Germany of the company with which we cooperated (*N* = 113,700), our study sample showed an identical mean age but a lower proportion of women (33% instead of 49%). In comparison to data from the working population of the German census and COSMO [54,88], our study population was slightly younger and better educated and had more men and fewer part-time workers. With regard to selection bias, it should be noted that many employees with already-positive attitudes toward prevention infection control measures participated. Compared to other studies focusing on differences between industries [73,89], we investigated the employees of one large company with a well-established safety culture and focused on differences between occupational groups. Therefore, the generalisability of our findings is limited particularly when comparing our findings to other occupational groups and smaller companies or to companies outside Germany. Furthermore, more than 97% of our participants were German citizens; hence, it was not possible to explore the influence of cultural aspects in detail. The major strength of our study was the rapid realisation of the investigation which enabled our survey to be carried out in the early stages of the pandemic. Due to the cross-sectional nature of this data analysis of baseline data, we cannot provide information on a causal relationship. However, we repeated the survey throughout the course of the pandemic within the same company. Therefore, we can evaluate changes in the individuals’ attitudes over time in future longitudinal analyses. We additionally distributed the same survey in other German companies and universities [33], to contrast our findings with findings in other employee samples. With regard to the mixed-methods approach we used throughout the project, the first results of the interviews conducted seem to confirm the findings of our employee survey from the leaders’ perspectives [84].

### 4.5. Meaning of the Study, Implications, and Future Research

The here presented cross-sectional findings from the beginning of the pandemic lead us to the assumption that the perceived probability of SARS-CoV-2-infection, attitude, and reactance toward OSH-measures for infection control differ between occupational groups. According to the results of our study, among other factors, the provided information about potential risks of infection with SARS-CoV-2 in the workplace and the affective risk perception seemed to be key parameters for positive attitudes and low reactance towards occupational safety and health measures for infection control. To counteract any potential for conflict and to promote compliance with the introduced infection control measures in the workplace, it seems to be important to consider occupational-group-specific context factors when implementing OSH-measures for infection control. Positive attitudes might strengthen the successful implementation and acceptance of OSH-measures. If employees, leaders, and colleagues all value and support those preventive OSH-measures, and if the workplace characteristics make it easy to follow those measures, then the employees are likely to develop the strong intention to carry out this behaviour in the long run [22]. Here, we presented findings from the cross-sectional analysis of baseline data. We aim to complement these cross-sectional findings conducting a longitudinal analysis where we address the impact of adjusted working conditions on attitudes toward OSH-measures for infection control over the course of the pandemic. Analysing follow-up data will further allow us to evaluate potential within-person and between-group changes regarding adjusted working conditions and alterations in attitudes.

## 5. Conclusions

The perceived probability of contracting COVID-19 in the workplace, attitudes toward technical, organisational, and personal OSH-measures for infection control, and resulting reactance, differ between remote and on-site working office and assembly line/manufacturing employees. Our findings provide a starting point for the occupational-group-specific implementation of OSH-measures in the workplace and expand the level of knowledge in attitudes toward infection control measures in the occupational setting. Occupational-group-specific contextual factors should be considered when implementing OSH-measures for infection control to promote employees’ success in transferring their positive attitudes toward OSH-measures into protective behaviours in the workplace and therefore maintain the workforce’s physical wellbeing during the COVID-19 pandemic.

## Figures and Tables

**Figure 2 ijerph-20-00614-f002:**
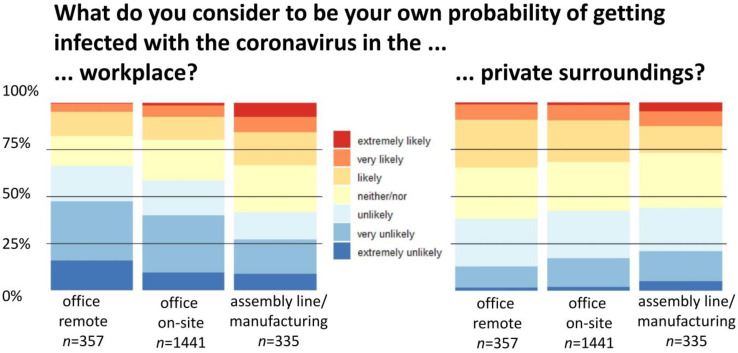
Perceived probability of contracting COVID-19 in the workplace and in private surroundings, compared by occupational group: office predominantly remote (*n* = 357), office on-site (*n* = 1441), and assembly line/manufacturing (*n* = 335).

**Figure 3 ijerph-20-00614-f003:**
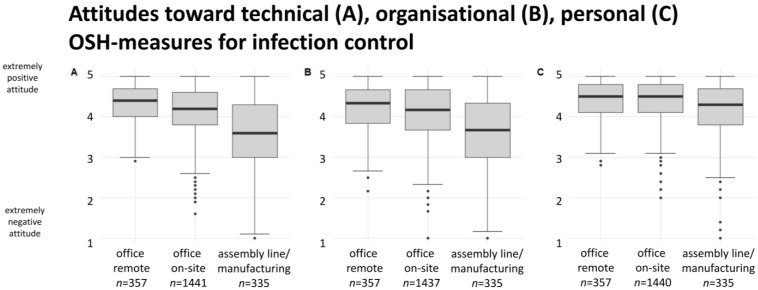
Boxplots showing the distribution of the assessed attitudes toward technical (**A**), organisational (**B**), and personal (**C**) OSH-measures for infection control, compared by occupational group: office remote, office on-site, and assembly line/manufacturing.

**Table 1 ijerph-20-00614-t001:** Socio-demographic and work-related characteristics at baseline: overall, and compared between predominantly remotely working office, on-site working office, and assembly line/manufacturing employees.

Characteristics	Overall	Office Remote	Office On-Site	Assembly Line/Manufacturing
*N* (%)	2144 (100%)	358 (16.7%)	1451 (67.7%)	335 (15.6%)
Gender (*n*) male female	2135 1444 (67.6%) 691 (33.4%)	358 204 (57.0%) 154 (43.0%)	1442 1003 (69.6%) 439 (30.4%)	335 237 (70.7%) 98 (29.3%)
Age in years (*n*) Mean ± SD IQR 18–29 years 30–59 years 60–67 years	2086 43.9 ± 11.1 (36.0–54.0) 271 (12.9%) 1697 (81.4%) 118 (5.7%)	350 44.0 ± 10.7 (35.0–52.0) 35 (10.0%) 285 (81.4%) 30 (8.6%)	1411 43.9 ± 11.0 (35.0–53.0) 183 (13.0%) 1155 (81.8%) 73 (5.2%)	325 43.6 ± 11.9 (35.0–53.0) 53 (16.6%) 257 (78.8%) 15 (4.6%)
Education [51] (*n*) primary intermediate high	2141 250 (11.7%) 692 (32.3%) 1199 (56.0%)	358 4 (1.1%) 69 (19.3%) 285 (79.6%)	1439 117 (8.1%) 426 (29.6%) 905 (62.9%)	335 129 (38.4%) 197 (58.9%) 9 (2.7%)
Affiliation to risk group [52] (*n*) yes no	2076 370 (17.8%) 1706 (82.2%)	345 69 (20.0%) 276 (80.0%)	1396 226 (16.2%) 1179 (84.5%)	326 75 (23.2%) 251 (76.8%)
Shift-work (*n*) yes no	2134 238 (11.1%) 1896 (88.8%)	357 0 357 (100%)	1434 45 (3.1%) 1398 (96.9%)	334 193 (57.6%) 141 (42.4%)
Provided information about potential risks of infection with SARS-CoV-2 in the workplace (*n*) Mean ± SD IQR	2119 4.3 ± 0.9 (4.0–5.0)	357 4.3 ± 0.9 (4.0–5.0)	1430 4.3 ± 0.8 (4.0–5.0)	323 3.8 ± 1.0 (3.0–4.0)
Trust in colleagues to adhere to distance rules (*n*) Mean ± SD IQR	2139 5.2 ± 1.4 (4.0–6.0)	357 5.3 ± 1.3 (5.0–6.0)	1440 5.3 ± 1.3 (5.0–6.0)	321 4.6 ± 1.7 (3.0–6.0)
COVID-19-specific resilience (*n*) Mean ± SD IQR	2135 5.4 ± 1.0 (4.8–6.0)	358 5.5 ± 0.9 (5.0–6.3)	1435 5.4 ± 0.9 (5.0–6.0)	333 4.9 ± 1.2 (4.3–5.8)
Affective risk perception (*n*) Mean ± SD IQR	2140 4.5 ± 1.2 (3.7–5.3)	358 4.6 ± 1.2 (4.0–5.3)	1439 4.5 ± 1.2 (3.7–5.3)	334 4.4 ± 1.5 (3.7–5.7)
Perceived probability of contracting COVID-19 in the workplace (*n*) Mean ± SD IQR	2144 3.3 ± 1.5 (2.0–4.0)	357 3.0 ± 1.5 (2.0–4.0)	1441 3.2 ± 1.5 (2.0–4.0)	335 3.8 ± 1.7 (2.0–5.0)
Perceived probability of contracting COVID-19 in private surroundings (*n*) Mean ± SD IQR	2142 3.8 ± 1.3 (3.0–5.0)	357 3.9 ± 1.2 (3.0–5.0)	1441 3.8 ± 1.3 (3.0–5.0)	335 3.8 ± 1.5 (3.0–5.0)
Reactance elicited by occupational infection control measures (*n*) Mean ± SD IQR	2142 2.7 ± 1.6 (1.5–3.8)	358 2.5 ± 1.6 (1.3–3.3)	1440 2.7 ± 1.5 (1.5–3.5)	335 3.3 ± 1.8 (1.9–4.5)
Attitudes toward technical OSH-measures for infection control (*n*) Mean ± SD IQR	2142 4.1 ± 0.7 (3.7–4.6)	357 4.3 ± 0.5 (4.0–4.7)	1441 4.1 ± 0.6 (3.8–4.6)	335 3.6 ± 0.9 (3.0–4.3)
Attitudes toward organisational OSH-measures for infection control (*n*) Mean ± SD IQR	2138 4.0 ± 0.7 (3.7–4.7)	357 4.2 ± 0.6 (3.8–4.7)	1437 4.1 ± 0.7 (3.7–4.7)	335 3.6 ± 1.0 (3.0–4.3)
Attitudes toward personal OSH-measures for infection control (*n*) Mean ± SD IQR	2141 4.4 ± 0.5 (4.1–4.8)	357 4.4 ± 0.5 (4.1–4.8)	1440 4.4 ± 0.5 (4.1–4.8)	335 4.2 ± 0.6 (3.8–4.7)

**Table 2 ijerph-20-00614-t002:** Attitude toward technical OSH-measures for infection control.

Professional Activity		Office Remote (*n* = 336)			Office On-Site (*n* = 1347)			Assembly Line and Manufacturing (*n* = 311)		
Block	Variables	Coef.	SE	*p*-Value	Coef.	SE	*p*-Value	Coef.	SE	*p*-Value
(I) Socio-demographics	Gender (vs. male) female	0.14	0.06	0.012 **	0.09	0.03	0.006 **	−0.26	0.11	0.015 *
	Age (vs. 18–29 years) 30–59 years 60–67 years	0.09 0.04	0.09 0.12	0.298 0.736	0.03 0.18	0.04 0.08	0.419 0.019 *	−0.06 0.27	0.13 0.25	0.646 0.279
(II) General workplace characteristics	Shift work (vs. no) Yes	-	-	-	-	-	-	−0.18	0.10	0.061
(III) Perception regarding the pandemic-related impact in the workplace	Reactance	−0.08	0.02	<0.001 ***	−0.09	0.01	<0.001 ***	−0.08	0.03	0.003 **
	Provided information about risks of infection with SARS-CoV-2 in the workplace	0.07	0.03	0.024 *	0.09	0.02	<0.001 ***	0.14	0.05	0.005 **
	Trust in colleagues to adhere to distance rules	0.03	0.02	0.126	0.07	0.01	<0.001 ***	0.11	0.03	<0.001 ***
(IV) Employees’ attitude toward COVID-19 in general	COVID-19-specific resilience	0.09	0.03	0.001 **	0.09	0.02	<0.001 ***	0.14	0.04	<0.001 ***
	Affective risk perception	0.05	0.02	0.035 *	0.11	0.01	<0.001 ***	0.10	0.03	0.002 **
Log-like and DF		−206.2137 on 652 degrees			−1017.34 on 2674 degrees			−365.5378 on 602 degrees		

* *p* < 0.05, ** *p* < 0.01, *** *p* < 0.001; All models were controlled for social desirability, 7-day-incidence, company site, and educational background. No Hauck–Donner effect was found.

## Data Availability

The data analysed during the current study are not publicly available due to German national data protection regulations. They are available on reasonable individual requests from the corresponding author.

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
