# Peer review of "Assessing Differences in Attitudes toward Occupational Safety and Health Measures for Infection Control between Office and Assembly Line Employees during the COVID-19 Pandemic in Germany: A Cross-Sectional Analysis of Baseline Data from a Repeated Employee Survey"

_ijerph, 2022, doi:10.3390/ijerph20010614_

Round 1

Reviewer 1 Report

An exceedingly well written article that clearly outlines the aim of the study, methods, analysis and conclusions. Well done. No further comments or suggestions. 
---------------------------------------------------------------------------

Review report

Attitudes towards occupational safety and health measures for infection control among office vs. assembly line employees during the COVID-19 pandemic in Germany: a cross-sectional analysis of baseline data from a repeated employee survey.

Could not identify anything wrong with this article. It is well written, with a distinct introduction, the review of the literature is good, and the methodology is well described and well suited to this type of study. The results are also well described both in text and figures, and there is a good connection between the two. One can sometimes find that figures, tables and text are not sufficiently interconnected in poorly written articles, but that is not the case in this study. The authors have done a stellar job creating a smooth transition between these elements.  

The only shortcoming of the study is that it is of cross-sectional character which always limits the ability to draw representative conclusions, so it is nothing unique to this study.
There is always something more that could be said in either of the sections but to keep it within the limitations of a journal article I believe the authors have done an excellent job, and it was a very readable article that I would recommend for publication. 

Reviewer 2 Report

I thought it was a well written and interesting article. Please find my comments as follows:   ​-​More clearly address the limitations of how the samples are not representative of the country or culture.
Minor Spelling/grammar adjustments​.   Overall, a detailed literature analysis, with clear presentation of results and good discussion inclusive of limitation considerations. Thank you.

Reviewer 3 Report

The last two sections (4.5 and 5) are a bit short and could be developed a bit more; otherwise everything about this paper is quite adequate, and the paper is worth publishing.

Reviewer 4 Report

- Author should be revised the title of manuscript with the concise and reflex of your research objective.

- Author should be added the recommendation from your finding in Abstract.

- Author should be added the more detail of inclusion and exclusion criteria of participant in the method.

- Author should be rechecked the size and front of Figure 2 and 3.

- Author should be added the respond rate in the result part.

- Author should be highlighted the recommendation part from your finding in the Meaning of the study, implication, future research.

- Author should be recommend the policy development implementation from your result of big sample size.

Reviewer 5 Report

General feed back

The survey was conducted via online questionnaire delivered between 10 August and 25 October 2020 at six German company sites of a worldwide leading global supplier of technology and services based in the federal states of Bavaria, Baden-Wurttemberg, and Lower Saxony. This survey aimed measuring  possible differences across occupational groups on employees’ perceived risk of infection with SARS-CoV-2 in the workplace, attitudes toward health protection measures ad associate factors.

2,144  employees (32.4% women), mainly working by distance (n=358), at an on-site office (n=1451), and at assembly line/manufacturing (n=335) were surveyed

The study provided interesting information but some points need to be addressed.

Line 188-196: It is unclear how the indicators were measured. Each questions was assigned a value along a 5-point Likert scale? And all questions were summed up to create a linear indicator? This should be better explained in the methods. Moreover, the authors should include a copy of the questionnaire as a supplementary file accessible to readers.

Line 329: “Our present study results” is a bad expression

Lines 351-358: This is an important that should be discussed. In general, compliance with health protection measures tends to be lower in middle-low size companies, which are more likely to go through financial hardships and lack resources to comply with risk reduction measures. 

Lines 359. Personal positions against health protection measures should be considered more in depth. I recommend to mention a comment about two different attitudes profiles/attitudes emerged during the pandemic with respect to health protection measured against the spread of COVID-19, “eudaimonic” and “hedonic” (as discussed by this study which I recommend to cite. PMID: 35035377). Likewise a comment on conspiracy theories for those reluctant against rules is needed.

Lines 416-417: feeling of being well-informed about potential health risks. A comment on information available online, including the issue of fake news is essential.

Lines 473-487: The section “Meaning of the study, implication, future research” is somehow redundant and should be summarized and incorporated in Conclusions
